# Contrastive Learning for Inference in Dialogue

**Etsuko Ishii, Yan Xu, Bryan Wilie, Ziwei Ji, Holy Lovenia, Willy Chung, Pascale Fung**

The Hong Kong University of Science and Technology

`{eishii, yxucb, bwilie, zjiad}@connect.ust.hk, pascale@ust.hk`

## Abstract

Inference, especially those derived from inductive processes, is a crucial component in our conversation to complement the information implicitly or explicitly conveyed by a speaker. While recent large language models show remarkable advances in inference tasks, their performance in inductive reasoning, where not all information is present in the context, is far behind deductive reasoning. In this paper, we analyze the behavior of the models based on the task difficulty defined by the semantic information gap – which distinguishes inductive and deductive reasoning (Johnson-Laird, 1988, 1993). Our analysis reveals that the disparity in information between dialogue contexts and desired inferences poses a significant challenge to the inductive inference process. To mitigate this information gap, we investigate a contrastive learning approach by feeding negative samples. Our experiments suggest negative samples help models understand what is wrong and improve their inference generations. [1]

## 1 Introduction

In conversations, inference is essential to uncover what the speaker intended to deliver, which often goes beyond the information explicitly expressed (Rieger, 1974; Thorndyke, 1976). Inferences can be made by an explicit or implicit logical reasoning based on utterances and common ground among speakers (Clark, 1975). By reading between the lines, these inferences enable appropriate responses in dialogues. This inference process has been intensely discussed in the early age of research at dialogues (e.g., Thorndyke, 1976). However, research in dialogue systems nowadays often overlook such an aspect and instead rely solely on the capabilities of large language models (LLMs) to understand and comprehend dialogues.

| | |
|---|---|
| Dial. | User A: I'm hungry, let's order up something to eat.
User B: Ok, maybe we can order a soup and a salad from the restaurant down the street.
User A: I was thinking of getting a hamburger, fries, and a chocolate sundae.
User B: You eat too much junk food. That sort of stuff clogs up your arteries and is very high in cholesterol.
User A: Well, I never seem to gain weight, so I don't mind.
User B: It's not only about getting fat or not, it's about being healthy. You could really have some health problems later on. *[Target]*
User A: How about pizza or maybe some fried chicken? Better yet , let's order some hot dogs!
User B: You are a lost cause. |
| Ques. | What is or could be the prerequisite of the target? |
| Gold | The speaker is a fitness freak and keeps track of his daily diet. |
| T5-base | The speaker eats too much junk food as it clogs up his arteries and is very high in cholesterol. |
| Ours | The speaker is a health-conscious person. |

Table 1: One example in "Conceivable" difficulty level, comparing the generated inferences from our method, T5-base, and the gold inference. *Dial.* and *Ques.* are short for *Dialogue* and *Question*. The snippets of inferences highlighted in pink are not explicitly stated in the dialogue and require the model to conduct inference inductively. We refer to this phenomenon as the *"information gap"* to accomplish this task.

Current LLMs, such as ChatGPT (OpenAI, 2022), lack the so-called "inductive reasoning" ability, while tending to accomplish the reasoning tasks deductively (Bang et al., 2023). It might be due to the fundamental difference between inductive and deductive processes. According to (Johnson-Laird, 1988, 1993), inductive reasoning involves an increase in semantic information from input to output while it remains the same in deductive reasoning. In the context of dialogue inference processes, especially when reading implicit messages, there are information gaps that need to be filled. For instance, somebody's invitation for a "quick

---

[1] The code and annotated data is available at `https://github.com/HLTCHKUST/contrastive_inference_dialogue`.

lunch as always" might be enough to specify the location and time without further interaction.

In this paper, we inspect the semantic information gap between dialogue contexts and intended inferences using a recently introduced dataset designed for generating inferences in dialogue (Ghosal et al., 2022). We hypothesize that the difficulty of the task can be associated with the amount of information gap required to bridge. We manually annotate the randomly sampled subset of the dataset regarding their information gap, and assess the performance of the models. The analysis shows a decline in model performance as the information gap increases.

Furthermore, we propose to apply a contrastive learning approach to improve inference performance. One limitation of the current sequence-to-sequence training, especially for reasoning tasks, is that models are never exposed to negative samples (Lee et al., 2021). In deductive reasoning, all the information required to generate an output is provided in the input, and there is no information gap. However, inductive reasoning requires including something that may not be explicitly stated in the input, and that is not simply learnable by only exposing gold samples. Thus, we need to teach the model with more guidance on the reasoning path. In our preliminary experiment using the same dataset and a multiple-choice framework with Roberta-large (Liu et al., 2019), we observed a significant improvement from an F1 score of 83.91 to 96.6 simply by feeding negative samples together with the other candidate, which indicates that feeding negative samples will help the model learn how to fill the information gap. Building on this initial experiment, our experimental results in the generative settings show that contrastive learning helps improve both overall and breakdown performance in each task difficulty level, especially for fully deductive and inductive cases. Additionally, we explore various sampling methods for generating negative samples for contrastive learning.

Our contributions are three-fold: (1) we provide data annotation based on the information gap and the assessment; (2) we suggest that the information gap accounts for the difficulty of the inference generation in dialogue; and (3) our experimental results show that the contrastive learning approach helps to fill the information gap.

## 2 Related Work

### 2.1 Inference in Conversation

As one of the most fundamental forms of the use of natural language (Jurafsky and Marin, 2023), advance in inference in conversation has been inseparable from the flourish of the field of natural language processing (NLP) (e.g., Mann, 1979; Phillips, 1975). Initially, the research focus of inference in conversation was to uncover the underlying rules of human conversations (e.g., Grosz, 1978; Carbonell Jr, 1978; Morgan, 1978). While it remains a core research question, recent works tend to be formed in question answering (QA) style so that we can test models in a handier way. Thanks to the powerful deep learning models, we can perform inference tasks sufficiently well yet leave underlying rules unclear. Recently, a number of QA datasets in conversational formats have been introduced (Choi et al., 2018; Reddy et al., 2019; Ma et al., 2018), and their main focus tends to be comprehension of non-conversational texts. To evaluate the comprehension of dialogues, various tasks have been proposed in different task formulations such as span extraction (Li et al., 2020; Yang and Choi, 2019; Wu et al., 2022), multiple choice (Sun et al., 2019), next utterance prediction (Cui et al., 2020), or natural language inference (NLI) (Welleck et al., 2019). Some tasks focus on a specific aspect of conversational inference, such as speaker guessing (Sang et al., 2022), and temporal reasoning (Qin et al., 2021). In natural language generation format, Ghosal et al. (2021, 2022) presents datasets for generating inferences based on dialogue, while Ghosal et al. (2021) only contains overt inferences and Ghosal et al. (2022) contains implicit guesses as well.

### 2.2 Task Difficulty and Information Gap

Controlling the difficulty of tasks requires delicate tuning as it is crucial for further advance in NLP; too challenging or too easy tasks cannot facilitate the growth of the technology. A task becomes more challenging if we impose additional conditions, such as limiting the amount of data and computational power or adding modality or other languages. Recently, some work has investigated the specific task with controlled or annotated data. For example, Williams et al. (2022) annotates on inference types such as numerical or reference to see which type is the most challenging in NLI. Cui et al. (2023) limit the data to assess the models'

capability to properly understand what the word "respectively" refers to in NLI.

Discussing the task difficulty independent of the models' performance is non-trivial. Current assessment of the task difficulty tends to be inseparable from the performance comparison of the models (e.g., Bang et al., 2023). In this way, we can observe the models' strengths and weaknesses across different tasks, but there is still a lack of absolute difficulty rankings of the tasks. One possible way to discuss the difficulty in a model- or task-agnostic way might be based on the information gap, which is the core challenge in inductive reasoning (Johnson-Laird, 1988, 1993). It has been discussed as "given and new information" in (Clark and Haviland, 1974; Clark, 1975) as the foundation in conversations, but this concept can be extended to any tasks (McKeown, 1979). In this line of work, Rudinger et al. (2020) proposes an NLI task in which an inference can be shifted when there is new information offered. These days, not many works explicitly mention "information gap" (Hayashi, 2022). However, we still have the concept underlain. For example, QA datasets commonly contain some portion of unanswerable questions (e.g., Rajpurkar et al., 2018; Bajaj et al., 2016) with the context provided.

## 2.3 Contrastive learning in NLG

Contrastive learning teaches a model to embed similar data sample pairs are closer and disparate sample pairs stay apart (Chopra et al., 2005; Smith and Eisner, 2005). Not only in obtaining better representations of words (Mikolov et al., 2013) or sentences (Fang et al., 2020; Gao et al., 2021; Liu et al., 2021a), contrastive learning is reported to improve a wide range of NLP tasks (e.g., Li et al., 2022b; Klein and Nabi, 2020) including text generation tasks (e.g., Cai et al., 2020; Li et al., 2021; Liu et al., 2021b; Paranjape et al., 2021; Li et al., 2022a; Shu et al., 2021). The main motivation for applying contrastive learning for sequence-to-sequence text generation tasks is that it allows the model to be exposed to negative samples during training (Lee et al., 2021). Indeed, negative samples are generated by some rule-based perturbations (Shu et al., 2021) or machine-generated texts (Cao and Wang, 2021) such as entity-swap (Tang et al., 2022) are reported to be effective for faithful, less hallucinatory text generation.

## 3 Information Gap in Inference

While existing work focuses on improving the model performance on inference tasks with various methods, there is still a lack of in-depth investigation on the task itself and how the model behavior is changed with the improved results. To fill this gap, we first propose to connect task difficulty with the "information gap" between contexts and target inferences and classify the inference task difficulty into three levels. Then, we focus on the generative inference in dialogues with the CICERO dataset (Ghosal et al., 2022). We collect additional annotations to assess the task difficulty of a subset of samples for further analysis.

### 3.1 Preliminaries of the CICERO Dataset

We denote a dialogue dataset as $\{\mathcal{D}^n\}_{n=1}^N$, and a dialogue as $\mathcal{D}_I = \{U_i\}_{i=1}^I$, where $U_i$ is an utterance at turn $i$. Given an input $X = (\mathcal{D}_I, Q, U_t)$ where $Q$ is a question and $U_t \in \mathcal{D}_I$ is a target utterance, we aim to learn a model $f_\theta$ to generate a plausible inference $\tilde{A} = f_\theta(X)$.

CICERO dataset comes with five types of questions:

1. **Cause**: What is or could be the cause of the target utterance?

2. **Prerequisite**: What is or could be the prerequisite of target?

3. **Subsequent Event (SE)**: What subsequent event happens or could happen following the target?

4. **Motivation**: What is or could be the motivation of target?

5. **Reaction**: What is the possible emotional reaction of the listener in response to target?

For subsequent event category, it also offers a more challenging setting called **Subsequent Event Clipped (SE_Clipped)** where the dialogue is clipped until the target utterance: $\mathcal{D}_t = \{U_i\}_{i=1}^t$.

### 3.2 Task Difficulty of the CICERO dataset

The CICERO dataset provides commonsense inferences made by human annotators. According to the annotation instructions, generated answers must be grammatically correct and consistent with the dialogue, yet they can be overt or speculative depending on contextual scenarios (Ghosal et al., 2022). While treated equally, some question types seem significantly more challenging than others according to the results breakdown reported in Ghosal et al. (2022). For example, Motivation scores the

highest even though it only accounts for 14% of the training set.

Although the surface format of the task is unified and thus cannot distinguish at a glance, we can sense that they challenge different things. For example, SE can be executed simply by summarizing the utterances after the turn $t$, while SE_Clipped required to predict future sequences from the dialogue. The difficulty differs even among questions in the same question type. Some inferences can be derived simply by paraphrasing the utterances, while others require logical guessing to read between the lines. These differences boil down to the information gap between the answer $A$ and the dialogue $\mathcal{D}_I$. Here, we take an initial step to investigate the task difficulties systematically and define three levels of difficulty based on the amount of information in the answer covered by the dialogue: Sufficient, Likely, and Conceivable.

**Level 1: Sufficient**    All the information in the answer is available in the given dialogue. Since there is no information gap between inputs and outputs, questions at this level are the easiest to answer. For example, from the given dialogue context below, it is overt that User A will be available on Saturday morning for delivery.

| | |
|---|---|
| User A | Can you deliver it, please? |
| User B | Yes, it costs two pounds fifty. |
| User A | All right, can you deliver here on Saturday? |
| User B | Sure. Does morning work for you? |
| User A | Sounds good. |
| Question | What is the prerequisite of the target utterance? |
| Answer | User A will be available on Saturday morning. |

**Level 2: Likely**    Some pieces of information in the answer are not available or directly stated, but it is possible to guess by combining the clues in the dialogue. Questions at this level can be compared to multi-hop question answering tasks (Yang et al., 2018; Welbl et al., 2018; Inoue et al., 2020). There are arguably different degree of hops to derive an answer depending on the context (Kumar et al., 2019; Cheng et al., 2021), however, here we classify all the questions that requires some sort of "hop" over e.g., a knowledge graph (Speer et al., 2017; Sap et al., 2019; Hwang et al., 2021) regardless of the degree. For example in the dialogue below, we can guess that User B will check the car as per User A's request. To check the car, User B will likely try to turn on the engine.

| | |
|---|---|
| User A | Jim, could you do me a favor? |
| User B | Sure, what can I do for you? |
| User A | My car has a problem starting. Could you please take a look at it for me? |
| User B | Sure thing. |
| Question | What subsequent event happens following the target utterance? |
| Answer | User B *tries to turn on the car engine.* |

**Level 3: Conceivable**    The answer contains some pieces of information that are not stated in the dialogue, and there is no clear guidance for a "hop". The answer is plausible but hardly verifiable. Questions at this level are not easy even with certain knowledge sources provided and can be compared to check hallucinations in open-domain text generations (Ji et al., 2023). For example, in the dialogue below, Bob may be a brother of User B, and his occupation could be a radio journalist, which is a plausible reason to call Bob to ask about the fire at the factory. However, we cannot verify the answer as the dialogue lacks the evidence to guess the relationship between the speakers and Bob, nor his occupation.

| | |
|---|---|
| User A | There's been a fire at the factory. |
| User B | Are you sure? There is nothing in the newspaper about it. |
| User A | I just saw it on the 6 o'clock news. |
| User B | I will phone Bob. |
| User A | Yeah, he always knows what's going on. |
| Question | What is the prerequisite of the target utterance? |
| Answer | User B's *brother* Bob is *a radio journalist.* |

### 3.3 Human Assessment of the Difficulty

To the best of our knowledge, there is no absolute automatic metric to compare two pieces of text in terms of the amount of semantic information they contain. Here, we assess the difficulty of the task defined in Section 3.2 by human annotation. We randomly select 75 samples per question type (in total 450 samples) from the CICERO test set. In our annotation scheme, we assign two well-trained annotators per sample to give a difficulty-level label and the other one expert to double-check and finalize the label. In a few cases where the three annotators disagreed on the label, an additional expert is assigned for confirmation.

In Table 2, we summarize the annotated results and the T5-base (Raffel et al., 2020) performance of the same subset that is fine-tuned on the CICERO training set. The CICERO dataset has a balanced mixture of the three levels (sufficient: 34.2%, likely: 33.6%, conceivable: 32.2%), and the per-

| Difficulty | BLEU-2 | METEOR | ROUGE_L | CIDEr |
|---|---|---|---|---|
| Sufficient (34.2%) | 18.78 | 16.80 | 29.37 | 46.07 |
| Likely (33.6%) | 16.38 | 15.89 | 26.76 | 32.27 |
| Conceivable (32.2%) | 11.92 | 12.72 | 21.87 | 22.23 |

Table 2: The performance of the fine-tuned T5-base gets worse along with the decrease in the amount of information available in the dialogue.

formance of T5-base uniformly degraded with the decrease of the amount of available information. As reported in Table 3, different question types have different proportions of difficulty levels as anticipated. Although the proportion of likely and conceivable questions can explain the difference in T5-base performance to a certain extent, it does not have a simple correlation. It may be due to the difference in which kind of information is required to bridge the gap between the dialogue and the answer. For example, speakers' emotional reactions might be easily guessed by the sentiment of the utterances, while identifying the cause of the utterance may involve a more complicated understanding of background knowledge.

## 4 Methodology

We primarily train our model $f_\theta$ by minimizing the negative log-likelihood:

$$\mathcal{L}_{\text{NLL}} = -\sum_{1 \leq n \leq N} \sum_{1 \leq j \leq k} \log p(a_j^n | a_{<j}^n, X^n),$$

where a generated inference is denoted as $\tilde{A}^n = \{a_j^n\}_{j=0}^k$. The contrastive learning objective is defined by:

$$\mathcal{L}_{\text{CL}} = -\sum_{1 \leq n \leq N} \log \frac{\exp(\text{sim}(\boldsymbol{h}_X, \boldsymbol{h}_{\tilde{A}^n})/\tau)}{\sum_{A' \in \mathcal{A}} \exp(\text{sim}(\boldsymbol{h}_X, \boldsymbol{h}_{A'})/\tau)},$$

where sim is a cosine similarity function, $\mathcal{A}$ is a set of negative samples of inferences, $\boldsymbol{h}_X$, $\boldsymbol{h}_{\tilde{A}^n}$, $\boldsymbol{h}_{A'}$ are the hidden representations of $X$, $\tilde{A}^n$, $A'$, and $\tau$ is a temperature, respectively. Following (Cao and Wang, 2021; Lee et al., 2021), the final training objective $\mathcal{L} = \mathcal{L}_{\text{NLL}} + \lambda\mathcal{L}_{\text{CL}}$, where $\lambda$ is a coefficient.

### 4.1 Selection of Negative Samples

Automatically generating a set of negative samples $\mathcal{A}$ for contrastive learning is a non-trivial task. The easiest method to sample negative samples is randomly sampling other inferences in the dataset (usually within the same batch), while the supervision of these negative samples might be weak due

to the dissimilarity of the sentences. We denote the contrastive loss for in-batch negative samples as $\lambda_{\text{b}}\mathcal{L}_{\text{CL}_{\text{b}}}$. Besides, we aim to feed more informative negative samples per gold inference, which we denote as $\lambda_{\text{s}}\mathcal{L}_{\text{CL}_{\text{s}}}$. Then, the training objective can be formed as $\mathcal{L} = \mathcal{L}_{\text{NLL}} + \lambda_{\text{b}}\mathcal{L}_{\text{CL}_{\text{b}}} + \lambda_{\text{s}}\mathcal{L}_{\text{CL}_{\text{s}}}$. Since the CICERO dataset also serves as an MCQ task, each inference has four high-quality plausible-looking yet not appropriate candidates. These counterfactual candidates are machine-generated and then filtered by human annotators. In our experiments, we explore the following ways for generating negative samples in fully-automatic:

**Non-Optimal Generation** Since the simple fine-tuning with $\mathcal{L}_{\text{NLL}}$ does not yield the optimal $f_\theta$ as reported in Table 3, we directly use generated inferences by the fine-tuned model. We use top-$k$ sampling with $k = 10$ for diversed generation.

**Replacement of Tokens** Inspired by (Park et al., 2021), we manipulate tokens of the gold inference using the prediction of a masked language model. More specifically, we compute the probability of each token in the gold inference $A$ when whole context $X$ and $A$ are given and when only $A$ is given. In this way, we can estimate which tokens in $A$ are more affected by the context $X$. We directly compare the log-likelihood score of each token and select tokens that differ more than a threshold. The selected tokens will be replaced by the randomly selected tokens in top-$k$ prediction by a masked language model. We apply the pretrained Roberta-large model (Replace$_{\text{ZS}}$) and the Roberta-large trained on the CICERO dataset for MCQ (Replace$_{\text{MCQ}}$), set $k = 10$, and the threshold 0.75.

## 5 Experiments

### 5.1 Baselines

We evaluate our proposed method across multiple Transformer-based models: T5-small/base/large (Raffel et al., 2020), and GPT2-base (Radford et al., 2019). To have a fair comparison, these baselines are finetuned on the CICERO training set only with $\mathcal{L}_{\text{NLL}}$. In addition, we compare our results with the performance of GPT-J (Wang and Komatsuzaki, 2021) and LLaMA-7B (Touvron et al., 2023) in a 3-shot setting. We report an average of three trials of randomly sample manually crafted prompts and

|  | | Difficulty | | | Automatic Metrics | | |
|---|---|---|---|---|---|---|---|
|  | Sufficient | Likely | Conceivable | BLEU-2 | METEOR | ROUGE_L | CIDEr |
| Cause | 46.7% | 33.3% | 20.0% | 11.93 | 13.78 | 21.88 | 34.65 |
| SE | 41.3% | 20.0% | 38.7% | 14.83 | 15.19 | 26.39 | 29.70 |
| SE_Clipped | 4.0% | 38.7% | 57.3% | 13.76 | 15.58 | 26.00 | 35.28 |
| Prerequisite | 32.0% | 28.0% | 40.0% | 6.77 | 10.19 | 15.82 | 12.31 |
| Motivation | 58.7% | 24.0% | 17.3% | 21.33 | 17.02 | 32.40 | 42.32 |
| Reaction | 22.7% | 57.3% | 20.0% | 23.30 | 18.72 | 33.96 | 35.75 |
| Total | 34.2% | 33.6% | 32.2% | 15.62 | 15.08 | 26.08 | 32.51 |

Table 3: The difficulty of the inferences varies on the type of questions, and so does the performance of the finetuned T5-base. The corresponding performance is calculated on the same subset of the CICERO test set.

|  | BLEU1 | BLEU2 | BLEU3 | BLEU4 | METEOR | ROUGE-L | CIDEr |
|---|---|---|---|---|---|---|---|
| GPT-J-3shot | 24.70 | 13.04 | 6.11 | 3.04 | 13.74 | 25.03 | 19.87 |
| + tf-idf | 22.83 | 11.64 | 5.52 | 2.83 | 12.19 | 22.45 | 17.99 |
| LLaMA-3shot | 28.34 | 15.18 | 7.25 | 3.73 | 15.26 | 27.43 | 26.47 |
| + tf-idf | 25.36 | 13.33 | 6.56 | 3.49 | 13.72 | 24.91 | 24.06 |
| T5-small | 29.20 | 15.66 | 8.19 | 4.67 | **15.88** | 27.34 | **33.58** |
| + CL | **29.46** | **15.83** | **8.29** | **4.71** | **15.88** | **27.63** | 33.44 |
| T5-base | 29.77 | 16.38 | 8.87 | 5.26 | 16.40 | 28.32 | 38.91 |
| + CL | **30.67** | **17.09** | **9.45** | **5.65** | **16.62** | **28.50** | **40.53** |
| T5-large | 29.57 | 16.79 | 9.45 | 5.81 | 16.60 | **29.06** | 43.38 |
| + CL | **30.07** | 17.02 | **9.56** | **5.83** | **16.67** | 28.90 | **43.80** |
| GPT2-base | 25.09 | 13.65 | 6.92 | 3.89 | 14.45 | 26.48 | 25.73 |
| + CL | **27.55** | **14.91** | **7.56** | **4.22** | **15.13** | **27.94** | **26.59** |

Table 4: Automatic results on CICERO test set. *CL* is short for ***contrastive learning***. We bold the better results between our method and the corresponding baseline model. We also highlight the best results across different models with underline.

| Plausibility | Win | Tie | Lose | $\kappa$ | Sufficient | | | Likely | | | Conceivable | | |
|---|---|---|---|---|---|---|---|---|---|---|---|---|---|
|  |  |  |  |  | Win | Tie | Lose | Win | Tie | Lose | Win | Tie | Lose |
| Ours vs T5-base | **38.7%**[*] | 35.8% | 25.5% | 0.73 | **45.7%**[*] | 34.6% | 19.7% | 34.7% | **40.4%** | 24.9% | **35.4%** | 32.4% | 32.2% |
| Ours vs Gold | 24.7% | **52.4%** | 22.9% | 0.21 | 22.3% | **53.0%** | 24.7% | 23.4% | **53.0%** | 23.6% | 28.5%[*] | **51.3%** | 20.2% |

Table 5: Human evaluation results on Plausibility, together with breakdown performance on each difficulty level. [*]Our model achieves a significant advantage over T5-base or Gold with pair-wise individual $t$-test ($p < 0.05$).

a strategic prompt using tf-idf to retrieve 3-most similar in-context examples.

## 5.2 Evaluation Metrics

**Automatic Metrics** In line with the CICERO paper, we assess the answers generated using $n$-gram overlap-based evaluation metrics: BLEU (Papineni et al., 2002), METEOR (Banerjee and Lavie, 2005), ROUGE-L (Lin, 2004), and CIDEr (Vedantam et al., 2015). Notably, CIDEr is calculated based on stem forms.

**Human Evaluation** For a comprehensive evaluation, we also conduct a human evaluation on *Plausibility* aspect which focuses on evaluating

whether the answers are rational or not. We evaluate the same data samples as those for task difficulty analysis. More specifically, comparing with both generated inferences from the T5-base model and the gold inferences. A/B testing is utilized to compare our proposed method and the corresponding baseline on the CICERO test set. Each comparison requires three judgments. The human evaluation is conducted based on a crowd-sourcing platform offered by Appen [2]. More details about human evaluation, such as annotator instructions and how the results are calculated, are included in Appendix A.2.

---

[2]https://client.appen.com/

| Model | BLEU-2 | METEOR | ROUGE_L | CIDEr |
|---|---|---|---|---|
| Ours | **17.09** | **16.62** | **28.50** | 40.53 |
| $-\mathcal{L}_{\mathrm{CL_s}}$ | 16.97 | 16.53 | 28.34 | **40.71** |
| $-\mathcal{L}_{\mathrm{CL_b}}$ | 16.95 | 16.53 | 28.49 | 40.18 |
| $-\mathcal{L}_{\mathrm{CL}}$ | 16.38 | 16.40 | 28.32 | 38.91 |

Table 6: Ablation study with the base model as T5-base.

## 5.3 Training Details

The models are trained using a batch size of 64 after gradient accumulation, with a learning rate set at $1\mathrm{e}{-}4$ for T5 models and $1\mathrm{e}{-}5$ for GPT-2 models. We limit the training to a maximum of 10 epochs, employing a linear learning rate scheduler. The checkpoint exhibiting the lowest perplexity on the validation set is chosen as the optimal model for each trial. In the case of contrastive learning, the temperature $\tau$ for $\mathcal{L}_{\mathrm{CL_b}}$ and $\mathcal{L}_{\mathrm{CL_s}}$ learning is set to 0.1 and 2.5, respectively, each contributing equally to the total loss with a coefficient $\lambda_{\mathrm{b}} = \lambda_{\mathrm{s}} = 0.5$. All the experiments are executed on a single RTX 3090 Ti GPU.

## 5.4 Results

We report the automatic results of both our method and the baselines in Table 4. Automatic metrics based on $n$-gram overlap are mostly improved thanks to contrastive learning. Moreover, our proposed method is model architecture-agnostic, given that it shows consistent improvement in different encoder-decoder T5 models and encoder-only GPT2. For GPT-J and LLaMA, we could not see any improvement introduced by tf-idf. We suspect that even though lexically similar, these examples may mislead the model to make wrong predictions

Overlap-based metrics can reflect the general quality of the generated inferences with respect to the gold answers. However, it does not reflect the inference ability of the generations, not to mention the inductive inference ability. In this work, we also explore the feasibility of NLI metrics for inference ability evaluation. More discussion is included in Section 6.6.

**Human Evaluation**  For a more comprehensive evaluation of inference ability, we conduct a human evaluation of the plausibility of the generated inferences and report in Table 5. We leverage pairwise individual $t$-tests to validate the significance of the improvements. Inter-annotator agreements

are computed using Fleiss' kappa ($\kappa$) [3] to assess the reliability of the evaluation. As it is shown in Table 5, contrastive learning significantly improves the plausibility of the generated inferences over T5-base with a substantial agreement. The generated inferences from T5-base with contrastive learning show comparable plausibility with gold ones in the CICERO test set with a fair inter-annotator agreement. The human evaluation further proves the effectiveness of our proposed method in improving inference ability. We further investigate the improvement breakdown in each difficulty levels to further analyze the effect of contrastive learning in Section 6.5.

## 6 Discussion

### 6.1 Case Study

Table 1 illustrates one example in "Conceivable", comparing the generated inferences from our method, T5-base, and the gold inference. While T5-base tends to copy from the dialogue (highlighted in blue ), contrastive learning promotes the model to infer more rational information which is not stated in the context (highlighted in pink ). We include more examples in Appendix B.2.

### 6.2 Ablation Study

We perform an ablation study on our proposed method using T5-base as the foundational model. The effectiveness of our model is compared against those trained without the application of either $\mathcal{L}_{\mathrm{CL_s}}$, $\mathcal{L}_{\mathrm{CL_b}}$, or both $\mathcal{L}_{\mathrm{CL}} = \lambda'_{\mathrm{b}}\mathcal{L}_{\mathrm{CL_b}} + \lambda'_{\mathrm{s}}\mathcal{L}_{\mathrm{CL_s}}$. In Table 6, our proposed method, employing both contrastive losses, amplifies the performance. A model devoid of $\mathcal{L}_{\mathrm{CL_s}}$ surpasses our own in terms of CIDEr, yet our method achieves superior results across all other metrics. Furthermore, the impact of the different contrastive losses varies across the range of automated methods. While $\mathcal{L}_{\mathrm{CL_b}}$ exhibits minimal impact on ROUGE-L, it proves more effective for CIDEr. The most significant contribution to the ROUGE-L improvement is derived from $\mathcal{L}_{\mathrm{CL_s}}$.

### 6.3 Comparison of Sampling Methods

In addition to the negative samples provided by the CICERO dataset, three different fully-automated methods of generating negative samples are explored as stated in Section 4.1. We train the model

[3] https://www.statsmodels.org/stable/generated/statsmodels.stats.inter_rater.fleiss_kappa.html

|              | BLEU-2 | METEOR | ROUGE_L | CIDEr |
|--------------|--------|--------|---------|-------|
| T5-base      | 16.38  | 16.40  | 28.32   | 38.91 |
| Contradiction | **17.09** | **16.62** | **28.50** | **40.53** |
| Non-optimal  | 16.40  | 16.40  | 28.19   | 39.17 |
| Replace$_{ZS}$ | 16.39 | 16.36 | 28.44 | 40.09 |
| Replace$_{MCQ}$ | 16.48 | 16.42 | 28.45 | 39.42 |

Table 7: Comparison of different sampling methods for generating negative samples. All the models are implemented based on T5-base.

|           | BLEU-2 | METEOR | ROUGE_L | CIDEr |
|-----------|--------|--------|---------|-------|
| T5-base   | 16.38  | 16.40  | 28.32   | 38.91 |
| + $m = 1$ | 16.67  | 16.43  | 28.31   | 40.43 |
| + $m = 2$ | 16.81  | 16.53  | 28.54   | 40.69 |
| + $m = 3$ | 16.82  | 16.52  | 28.45   | **40.94** |
| + $m = 4$ | **17.09** | **16.62** | **28.50** | 40.53 |

Table 8: The effect of the amount of negative samples. We report the average of three trials for $m = 1, 2, 3$.

leveraging the negative samples obtained from different methods, respectively and present the performance of the models in Table 7 for comparison. While different generation methods yield different improvements in the automatic metrics, in general, feeding negative samples does not hurt taining the models to perform dialogue inference. The "contradiction" negative samples from the dataset provide the largest improvement to the model performance, which suggests that higher quality of negative samples can guide the models better with a smaller amount. Another method that shows effectiveness is to replace the words that affect the predictions largely for the RoBERTa-large model trained to differentiate positive samples from negative samples, Replace$_{MCQ}$, while replacement measured by the RoBERTa model in a zero-shot way (Replace$_{ZS}$) is less helpful. This indicates that fine-tuned RoBERTa assigns the probability of the tokens more informatively for inference. Our exploration of using a non-optimal T5-base model to generate negative samples is expected to improve the model performance by iterative self-contrasting. However, self-improvement may not be effective without further human filtering since we might include rational answers as negative samples, which introduce noise during training.

### 6.4 Effect of the Amount of Negative Samples

In our main experiments, we feed all four of the the counterfactual candidates provided by the CICERO dataset as negative samples to compute $\mathcal{L}_{CL_s}$. As

| Difficulty | BLEU-2 | METEOR | ROUGE_L | CIDEr |
|------------|--------|--------|---------|-------|
| Sufficient | 25.24 (**+6.46**) | 20.54 (**+3.73**) | 36.58 (**+7.21**) | 82.07 (**+36.00**) |
| Likely     | 19.32 (**+2.94**) | 17.98 (**+2.09**) | 30.80 (**+4.04**) | 46.14 (**+13.87**) |
| Conceiv.   | 17.09 (**+5.17**) | 15.39 (**+2.67**) | 28.62 (**+6.75**) | 38.99 (**+16.76**) |

Table 9: The performance is improved thanks to the contrastive learning across all the difficulty levels. *Conceiv.* is short for *Conceivable*. The performance is calculated on the same subset of the CICERO test set in Table 2.

the effective amount of negative samples for contrastive learning is under discussion (e.g., Awasthi et al., 2022; Nozawa and Sato, 2021), we conduct a control experiment by feeding randomly sampled counterfactual candidates ($m = 1, 2, 3$) to observe the effect of the number of negatives. We report the results in Table 8; note that we report the average of three trials with different random seeds for $m = 1, 2, 3$. The performance generally improves along with the increase in the number of negative samples, implying that the high-quality negative samples contribute to teach the model to inference. Encouraged by our results, it would be interesting to quantify how much guidance is necessary for each level. For example, the "Sufficient" level may need fewer negative samples than the "Conceivable" level to achieve similar performance. It would be also beneficial to investigate the possibility of dynamically controlling the number of negative samples to feed.

### 6.5 Analysis of Improvements based on Task Difficulty

We further investigate how contrastive learning improves the model performance in different task difficulties. Table 9 reports the automatic score breakdown based on the difficulty annotated. Compared to the performance of the T5-base model reported in Table 2, our method yields improvement for all the levels, especially on "Sufficient" and "Conceivable". Similarly, we list the breakdown of human evaluation results to each task difficulty level in Table 5. T5-base with contrastive learning outperforms T5-base on plausibility in all difficulty levels, especially for "Sufficient" and "Conceivable", which is consistent with the trend of automatic metrics. In the "Sufficient" level, the advantage of our model is significant over the T5-base. This proves that contrastive learning can effectively improve the model's inference ability. Moreover, our method even significantly wins over gold in the "Conceivable" level in human evaluation with $p < 0.05$. Conceivable level gold answers tend to

| | UNLI$_{gold}$ | UNLI$_{con}$ | AS$_{gold}$ | AS$_{con}$ |
|---|---|---|---|---|
| GOLD | 1.0000 | 0.5220 | 0.9995 | 0.3670 |
| T5-small | 0.2283 | 0.6584 | 0.0577 | 0.6224 |
| + CL | 0.2271 | 0.6422 | 0.0565 | 0.5940 |
| T5-base | 0.2607 | 0.6894 | 0.0765 | 0.6355 |
| + CL | 0.2572 | 0.6586 | 0.0760 | 0.6009 |
| T5-large | 0.2947 | 0.7015 | 0.0973 | 0.6282 |
| + CL | 0.2940 | 0.6993 | 0.0931 | 0.6106 |
| GPT2-base | 0.2783 | 0.6460 | 0.0783 | 0.5723 |
| + CL | 0.3066 | 0.6610 | 0.0964 | 0.5561 |

Table 10: NLI-based metric results on the CICERO test set. AS is short for *AlignScore*.

include something that is not stated/verifiable in the dialogue contexts provided, while ours tends to be more supported by the dialogue context (see Table 1). We believe this resulted in ours being more favored by annotators.

## 6.6 Challenges in Evaluation of Inductive Reasoning

As we have discussed in the previous sections, it is extremely challenging to evaluate inductive processes because, by nature, outputs contain new information that is not stated in the inputs (Johnson-Laird, 1988, 1993). While the field has been aware of the fundamental difference between inductive and deductive for more than 60 years (Watanabe, 1960), there is no way to directly compare two pieces of text in terms of the amount of "semantic information" until now. Recently, with the arising demand in faithful and factual text generation, several metrics have been applied mainly by computing overlap in named entities or extracted keywords (Mao et al., 2021). Although the overlap-based metrics could be a decent starting point for many tasks such as summarization, it is not appropriate for inference in dialogue as non-overlap is something desired rather than being avoided.

Another common choice to measure the plausibility today would be adopting NLI-based metrics (Honovich et al., 2022). In Table 10, we report model-based NLI metrics of UNLI (Chen et al., 2020) and AlignScore (Zha et al., 2023). We measure entailment between generated inferences and the gold references (UNLI$_{gold}$/AS$_{gold}$), or between generated inferences and the corresponding dialogue context (UNLI$_{con}$/AS$_{con}$) on a scale of $[0, 1]$. The training specifics of the NLI models, as well as their performance, can be found in Appendix A.1.

Despite being promising, the NLI scores are

hardly interpretable, showing the consistent trend of contrastive learning degrading except for the GPT2-base. Even gold answers are labeled as "neutral" and undeterminable, and it is difficult to associate numbers with the quality of generated inferences. Although NLI metrics are an effective method to quantify factuality (Zha et al., 2023), this result suggests that NLI metrics are not suitable for inference in dialogue. Future work is needed to investigate possible evaluation metrics of the information gap since it can also benefit a wide range of NLP tasks.

## 7 Conclusion

In this paper, we conduct an analysis of inferences in dialogue, focusing on the availability of semantic information between inputs and outputs. As expected, the models perform worse on the samples with larger information gaps. We investigate a contrastive learning approach to teach what is wrong in inference. Our experimental results suggest the effectiveness of our approach, showing the promising direction to bridge the information gap, especially for smaller models with <1B parameters.

## Limitations

The main drawback of the proposed method is that it requires more computational resources and longer training time as we increase the amount of training data to yield improvement with contrastive learning over the baselines. Although our method is model, dataset, and language agnostic, our exploration is limited to the popular transformer-based architectures and the single dataset in English.

The other significant aspect we have not covered in the paper (and in most of the literature to the best of our knowledge) is the stopping rule of the inference process in dialogue. As suggested in Clark (1975), there is a clear boundary between what portion of the untold information should be guessed and what can be left unknown in a speaker's intention. However, even in dataset construction phases, this aspect has been neglected (e.g., Bhagavatula et al., 2020; Ghosal et al., 2022). The stopping rule is essential since it can be one factor separating "Likely" questions and "Conceivable" questions. An important question for future studies is how to deal with the stopping rule, as it can be also associated with a boundary of hallucination and acceptable freedom in open-domain dialogue systems.

## Ethics Statement

In this work, we collect additional human annotations on the task difficulty in the CICERO dataset. The CICERO dataset is publicly available, and we will also release our task difficulty annotations upon acceptance. We consider the target inference provided by the dataset as the gold inference and analyze the task difficulty fully based on the relationship between the target inferences and the dialogues. We conduct human evaluation relying on a public crowd-sourcing platform, Appen. Each judgment takes four seconds on average, and we assign 15 cents as the payment to each judgment for annotators. No personal information except the country-level location is used to ensure the English proficiency of the annotators to guarantee the annotation quality, while all annotations were anonymized before the analysis.

## Acknowledgement

The authors thank all the anonymous reviewers for their valuable comments and constructive feedback. This work has been partially supported by China NSFC Project (No. NSFC21EG14), the Hong Kong Jockey Club (RG192/HKJCCT21EG01), and the Hong Kong PhD Fellowship Scheme, Research Grant Council, Hong Kong (PF18-25016).

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

## A   Additional Details of Experiments

### A.1   Details of NLI Models

**UNLI Model**   Following Chen et al. (2020), we apply RoBERTa-large model (Liu et al., 2019) on the $u$-SNLI datasets. We first train the model on ANLI dataset which is to classify into three categories {entail, neutral, contradict } on SNLI (Bowman et al., 2015), MNLI (Williams et al., 2018), FEVER-NLI (Thorne et al., 2018), ANLI (R1, R2, R3) (Nie et al., 2020), and then switch the classifier to be regression way and train on $u$-SNLI datasets. [A.1]   In our observation, this warmup helps to improve the performance on $u$-SNLI. Our training batch size is 16, and we train for 3 epochs with the learning of $1e-5$. In Table A.1, we report the scores on $u$-SNLI development set and test set along with the numbers reported in Chen et al. (2020): the Person's correlation coefficient $r$, the Spearman rank correlation $\rho$, and the mean square error (MSE). In our main experiments, we use the best model as the model-based evaluation metric.

**AlignScore Model**   Following Zha et al. (2023), we apply the RoBERTa-large model, and the checkpoint distributed by the authors[A.2].

### A.2   Details of Human Evaluation

As it is stated in Section 5.2, a human evaluation on *Plausibility* is conducted on 450 data samples in total across six subtasks from the CICERO test set. We compare the generated inferences from our model with those from T5-base and gold inferences. A/B testing is utilized and we ensure that each data sample are evaluated by three different annotators. For quality control, we limit the annotators' locations to the United States, United Kingdom, Canada, or Australia to ensure English proficiency. All the annotators are required to answer 20 test questions with more than 80% accuracy before starting the annotation. During the human evaluation, we present the same context and two options from different models to annotators in comparison. The annotators are required to decide which inference is more plausible by choosing from "Option 1", "Option 2", "both", and "neither". The annotator instruction is presented in Figure A.1.

After collecting all the annotations, we first calculate the ratio of Win/Tie/Loss of our model with respect to T5-base and Gold, respectively. The corresponding results are shown in Table 5. Moreover, the inter-annotator agreement is also calculated based on Fleiss Kappa. We implement Fleiss Kappa ($\kappa$) based on the "statsmodels" package [A.3]. To calculate the significance level of the advantage of our method over the baseline and gold, we also calculate the winning rate of each model under evaluation in Table A.2. More specifically, the model will gain one score if the annotator chooses the corresponding option or "both", or the model will gain a zero score instead. We take an average over all the scores and consider that to be the human evaluation result of the model. For better representation, we show these results as percentages. We calculate the significance level with pair-wise individual t-tests given the scores from all the samples.

### A.3   Choice of $\lambda_b$ and $\lambda_s$

The coefficient $\lambda_b$ and $\lambda_s$ is set to be $\lambda_b = \lambda_s = 0.5$ based on the preliminary experiments reported in Table A.3.

## B   Additional Experimental Results

### B.1   Score breakdown of question types

In Table B.1, we report the breakdown of the automatic evaluation results of each question type of the CICERO test set.

### B.2   Case Study

In Table 1, we show one example in the "Conceivable" level. We also present the examples in the "Sufficient" and "Likely" levels in Table B.3 and Table B.2, respectively.

---

[A.1] https://huggingface.co/ynie/roberta-large-snli_mnli_fever_anli_R1_R2_R3-nli
[A.2] https://github.com/yuh-zha/AlignScore
[A.3] https://www.statsmodels.org/stable/index.html

| Roberta-large | Dev | | | Test | | |
|---|---|---|---|---|---|---|
| | $r$ | $\rho$ | MSE | $r$ | $\rho$ | MSE |
| Chen et al. (2020) | 0.6383 | 0.6408 | 0.0751 | 0.6271 | 0.6346 | 0.0777 |
| Ours | **0.7334** | **0.7499** | **0.0635** | **0.7369** | **0.7525** | **0.0643** |
| Ours w/o warmup | 0.7115 | 0.7209 | 0.0666 | 0.7226 | 0.7370 | 0.0655 |

Table A.1: The performance on the $u$-SNLI development and test set.

| Model | | Plausibility | Sufficient | Likely | Conceivable |
|---|---|---|---|---|---|
| Our vs T5-base | Ours | **47.5%** | **53.7%** | 45.0% | 43.4% |
| | T5-base | 34.3% | 27.7% | 35.3% | 40.2% |
| Ours vs Gold | Ours | 35.1% | 35.3% | 30.9% | **39.3%** |
| | Gold | 33.3% | 37.7% | 31.1% | 31.0% |

Table A.2: Human evaluation results calculated based on the winning rate of each model. The results in bold are significantly better than that for the other model with pair-wise individual t-tests ($p < 0.05$).

| $\lambda_{\text{b}}$ | $\lambda_{\text{s}}$ | BLEU-1 | BLEU-2 | BLEU-3 | BLEU-4 | METEOR | ROUGE_L | CIDEr |
|---|---|---|---|---|---|---|---|---|
| 0.5 | 0.5 | **30.67** | **17.09** | **9.45** | **5.65** | **16.62** | **28.50** | **40.53** |
| 1.0 | 1.0 | 30.29 | 16.69 | 9.09 | 5.42 | 16.55 | 28.31 | 38.77 |
| 2.0 | 2.0 | 29.81 | 16.37 | 8.88 | 5.28 | 16.32 | 28.05 | 38.18 |

Table A.3: Automatic results on CICERO test set with different $\lambda_{\text{b}}$ and $\lambda_{\text{s}}$.

# Which Answer Is More Plausible?

Instructions ▲

## Overview

In this job, you will be presented with a conversation between user A and user B, and a question on the conversation. Review the conversation carefully to determine which option is **more plausible**.

### Steps

1. Read the conversation and the question.
2. Choose more plausible answer.

### Examples

| **Dialogue 1** | **Plausible: Option 1** |
| --- | --- |
| User A: Can you deliver it, on Saturday?
User B: Sure. Does morning work for you?
User A: Sounds good.
**Question:** What is the prerequisite of the target utterance?
**Option 1:** User A will be available on Saturday morning.
**Option 2:** User B does not knows User A's home address. | Reason: Option 1 answers the question correctly using the information stated in the provided dialogue while Option 2 seems contradicting to the conversation. |
| **Dialogue 2** | **Plausible: Both** |
| User A: Jim, my car has a problem. Could you please take a look at it for me?
User B: Sure thing.
**Question:** What subsequent event happens following the target utterance?
**Option 1:** User B tries to turn on the car engine.
**Option 2:** User A and User B go together to the parking lot. | Reason: Both Option 1 and Option 2 seem plausible based on the dialogue. |
| **Dialogue 3** | **Plausible: Neither** |
| User A: There's been a fire at the factory.
User B: Are you sure? There is nothing in the newspaper about it. I will phone Bob.
User A: Yeah, he always knows what's going on.
**Question:** What is the prerequisite of the target utterance?
**Option 1:** User B's brother Bob works in a restaurant nearby.
**Option 2:** User A was annoyed by Bob being gossipy. | Reason: While both Option 1 and Option 2 might be possible, there is no clear clue to choose neither of the option. |

---

Read the dialogue below and assess the plausibility of the options on the question regarding the target utterance:

Dialogue:

User A: Hi, Mike! How are you feeling now?
User B: How did you know I was here? Is it Tom?
User A: I was talking with Bob yesterday and I learnt your right leg had been injured. How did it happen?
User B: Their right back Tom knocked me down when I rushed to their goal with the ball.
User A: Wow! He must have hit you hard.
User B: Of course. He hit me from the back and sent me rolling over and over. At the time I had a lot of pain. Anyway, they brought me here.
User A: Nothing serious, I hope.
User B: The doctor said there weren't any internal injuries, but that I'd better stay here a couple of days.
User A: Well, Mike. Take it easy.
User B: Thank you for your coming. And thanks for the flowers.

Target: Their right back Tom knocked me down when I rushed to their goal with the ball.

Question: What is or could be the cause of target?

Option 1: Tom hit mike from the back and sent him rolling over and over.

Option 2: The listener inquired about the speaker's injury.

**Which answer seems more plausible?** (required)
○ Option 1
○ Option 2
○ Both
○ Neither

Figure A.1: Annotator instruction of the human evaluation on *Plausibility*.

| | | BLEU-1 | BLEU-2 | BLEU-3 | BLEU-4 | METEOR | ROUGE_L | CIDEr |
|---|---|---|---|---|---|---|---|---|
| | LLaMA | 24.72 | 12.55 | 5.87 | 3.02 | 13.78 | 23.94 | 26.11 |
| | T5-small | 27.21 | 13.54 | 7.06 | 4.24 | 15.56 | 24.88 | 36.89 |
| | + CL | 27.60 | 13.70 | 6.94 | 4.06 | 15.61 | 25.09 | 35.36 |
| | T5-base | 27.65 | 14.52 | 7.92 | 4.91 | 16.13 | 25.80 | 43.37 |
| Cause | + CL | 29.15 | 15.35 | 8.61 | 5.36 | 16.65 | 26.32 | 45.67 |
| | T5-large | 27.79 | 14.87 | 8.27 | 5.19 | 16.26 | 26.29 | 46.81 |
| | + CL | 27.64 | 15.03 | 8.44 | 5.28 | 16.32 | 26.11 | 48.40 |
| | GPT2-base | 21.51 | 11.05 | 5.33 | 3.04 | 13.52 | 23.32 | 24.55 |
| | + CL | 18.67 | 9.95 | 4.83 | 2.67 | 12.85 | 22.81 | 23.08 |
| | LLaMA | 30.99 | 15.43 | 6.76 | 3.43 | 15.76 | 28.12 | 26.61 |
| | T5-small | 30.27 | 14.88 | 6.52 | 3.43 | 15.85 | 26.87 | 28.14 |
| | + CL | 30.75 | 15.12 | 6.80 | 3.64 | 15.90 | 27.32 | 28.38 |
| | T5-base | 30.92 | 15.70 | 7.44 | 4.18 | 16.31 | 27.91 | 33.54 |
| SE | + CL | 31.19 | 16.17 | 8.01 | 4.58 | 16.51 | 27.90 | 34.95 |
| | T5-large | 30.65 | 16.18 | 8.20 | 4.82 | 16.45 | 28.62 | 36.56 |
| | + CL | 31.26 | 16.37 | 8.26 | 4.79 | 16.56 | 28.65 | 37.30 |
| | GPT2-base | 26.85 | 13.67 | 6.02 | 3.23 | 14.65 | 26.63 | 22.64 |
| | + CL | 26.10 | 13.25 | 5.95 | 3.18 | 14.61 | 27.49 | 23.98 |
| | LLaMA | 30.69 | 15.23 | 6.61 | 3.30 | 15.84 | 28.25 | 26.60 |
| | T5-small | 29.91 | 14.64 | 6.36 | 3.35 | 15.70 | 26.84 | 27.95 |
| | + CL | 30.18 | 14.84 | 6.58 | 3.47 | 15.73 | 27.20 | 27.79 |
| | T5-base | 30.39 | 15.24 | 7.05 | 3.85 | 16.08 | 27.67 | 31.14 |
| SE_Clipped | + CL | 31.06 | 15.92 | 7.65 | 4.25 | 16.21 | 27.68 | 32.45 |
| | T5-large | 30.09 | 15.73 | 7.80 | 4.47 | 16.17 | 28.49 | 35.24 |
| | + CL | 30.80 | 15.96 | 7.83 | 4.45 | 16.33 | 28.37 | 35.45 |
| | GPT2-base | 26.74 | 13.45 | 5.70 | 2.98 | 14.67 | 26.48 | 21.68 |
| | + CL | 26.55 | 13.34 | 5.81 | 3.04 | 14.67 | 27.44 | 22.33 |
| | LLaMA | 21.38 | 11.64 | 5.46 | 2.68 | 13.11 | 23.04 | 22.05 |
| | T5-small | 18.36 | 9.61 | 4.76 | 2.52 | 12.60 | 20.53 | 26.02 |
| | + CL | 18.49 | 9.81 | 4.78 | 2.41 | 12.48 | 20.79 | 26.37 |
| | T5-base | 19.32 | 10.21 | 5.03 | 2.64 | 13.15 | 21.46 | 29.89 |
| Prerequisite | + CL | 20.04 | 10.94 | 5.69 | 3.20 | 13.26 | 21.92 | 31.66 |
| | T5-large | 19.18 | 10.63 | 5.60 | 3.17 | 13.46 | 22.28 | 33.78 |
| | + CL | 19.75 | 10.87 | 5.80 | 3.27 | 13.53 | 21.92 | 34.13 |
| | GPT2-base | 15.73 | 8.75 | 4.39 | 2.37 | 11.35 | 20.32 | 20.53 |
| | + CL | 13.09 | 7.56 | 3.69 | 1.97 | 10.91 | 19.64 | 20.28 |
| | LLaMA | 31.22 | 20.65 | 11.90 | 6.43 | 17.36 | 32.88 | 36.77 |
| | T5-small | 34.26 | 24.25 | 16.56 | 10.51 | 18.73 | 36.10 | 56.23 |
| | + CL | 34.11 | 24.35 | 16.64 | 10.54 | 18.66 | 36.24 | 56.39 |
| | T5-base | 35.23 | 25.07 | 17.20 | 11.31 | 19.81 | 37.60 | 66.52 |
| Motivation | + CL | 35.74 | 25.53 | 17.59 | 11.67 | 19.97 | 38.00 | 68.46 |
| | T5-large | 34.94 | 25.38 | 17.83 | 12.29 | 20.52 | 38.83 | 77.02 |
| | + CL | 35.59 | 25.73 | 17.92 | 12.14 | 20.30 | 38.41 | 75.54 |
| | GPT2-base | 30.12 | 21.14 | 14.09 | 8.88 | 17.14 | 35.05 | 42.98 |
| | + CL | 29.05 | 20.73 | 14.14 | 9.01 | 17.34 | 35.85 | 43.77 |
| | LLaMA | 20.77 | 15.03 | 9.21 | 5.31 | 15.24 | 30.54 | 18.45 |
| | T5-small | 34.15 | 23.55 | 15.20 | 9.44 | 18.50 | 36.17 | 39.39 |
| | + CL | 33.51 | 23.16 | 15.01 | 9.46 | 18.26 | 36.10 | 39.96 |
| | T5-base | 33.98 | 23.73 | 15.62 | 10.10 | 19.05 | 37.18 | 45.98 |
| Reaction | + CL | 33.95 | 23.81 | 15.68 | 10.04 | 19.08 | 37.04 | 46.78 |
| | T5-large | 33.26 | 23.67 | 15.92 | 10.53 | 19.34 | 37.52 | 55.21 |
| | + CL | 34.51 | 24.44 | 16.37 | 10.80 | 19.35 | 37.23 | 54.82 |
| | GPT2-base | 26.15 | 16.89 | 10.73 | 6.54 | 16.17 | 32.20 | 29.90 |
| | + CL | 25.34 | 18.24 | 12.04 | 7.40 | 16.56 | 34.52 | 25.75 |

Table B.1: Automatic results on the CICERO test set for each question type.

| | |
|---|---|
| Dialogue | User A: Hello, what can I do for you? |
| | User B: Hello, I come to pay my water and electricity fees . |
| | User A: Give me your water and electricity bills, please. |
| | User B: Here they are. |
| | User A: You should pay 160 yuan for the electricity fee and 80 yuan for the water fee. |
| | User B: Do you mean that I should pay 240 yuan in total? *[Target]* |
| | User A: Yes. Will you pay by cash or credit card? |
| | User B: Cash, please. Here is the money. |
| | User A: I get 250 yuan from you, and this is the change, 10 yuan. |
| | User B: OK. Thank you. Bye-bye. |
| | User A: Bye . |
| Question | What is or could be the motivation of the target? |
| Gold | The speaker is shocked to hear the total amount to be paid for his utility bill. |
| T5-base | The speaker is curious to know if he should pay 240 yuan for electricity and 80 yuan for water fees. |
| Ours | The speaker is curious to know how much he has to pay for electricity and water . |

Table B.2: One example in "Likely" difficulty level, comparing the generated inferences from our method, T5-base, and the gold inference. We highlight the false inference in green .

| | *Example 1* |
|---|---|
| Dialogue | User A: Airports are sad places. |
| | User B: Sometimes, I guess. But we'll write to each other. You'll come down at Christmas. |
| | User A: If we can find the money. |
| | User B: Don't worry, Marta. Everything will be taken care of. They say that fares are going to be reduced in the next six months. |
| | And when I graduate, well ... |
| | User A: That's two years from now. Two years is a long time. |
| | User B: The time will pass quickly. You'll see. I might even be able to go back to New York next summer. |
| | User A: Oh, John, you'll forget all about me. Your mother will find you a nice girl, you'll get married, and live happily ever after. |
| | User B: No, I won't. I swear I won't. Believe me please. |
| | User A: Whatever you say, all I know is that you are going to be taken away from me. |
| | User B: That's ridiculous[ I'll write every day, whether you answer me or not. |
| | User A: Don't be silly. You'll have other things to do. (She begins to cry.) |
| | User B: Don't cry, Marta, please. *[Target]* |
| Question | What is the possible emotional reaction of the listener in response to the target? |
| Gold | The listener was sad that she wouldn't be able to see john for a long time. |
| T5-base | The listener is happy to help marta. |
| Ours | The listener is feeling sad for marta. |

| | *Example 2* |
|---|---|
| Dialogue | User A: Honey , I think you should quit smoking. |
| | User B: Why? You said I was hot when smoking. |
| | User A: But I want you to be fit. |
| | User B: Smoking is killing. I know. |
| | User A: Check out this article . It says smoking can lead to lung cancer. *[Target]* |
| | User B: I don't believe it. |
| | User A: But you know that smoking does harm to health, right? |
| | User B: Of course I know it, but you know it's hard to quit smoking ... |
| | User A: Stop beating around the bush. Will you quit or not? |
| | User B: Yes, ma'am. Whatever you say. |
| Question | What is or could be the prerequisite of the target? |
| Gold | The article in the magazine was about lung cancer caused by smoking. |
| T5-base | The speaker is a smoker . |
| Ours | The speaker has read an article about smoking . |

Table B.3: Two examples in "Sufficient" difficulty level, comparing the generated inferences from our method, T5-base, and the gold inference. We highlight the false inference in green .