# OpenReview forum: "Contrastive Learning for Inference in Dialogue"
_EMNLP/2023/Conference — EMNLP 2023 Main_

### Official Review · Reviewer_LNAb · 2023-08-02

**Soundness:** 3

**Excitement:**

3: Ambivalent: It has merits (e.g., it reports state-of-the-art results, the idea is nice), but there are key weaknesses (e.g., it describes incremental work), and it can significantly benefit from another round of revision. However, I won't object to accepting it if my co-reviewers champion it.

**Paper Topic And Main Contributions:**

This paper conducts an analysis of inferences in dialogue, focusing on the availability of semantic information between dialogue context and outputs.
The authors manually annotate three levels of task difficulty based on the amount of information in the answer covered by the dialogue in the CICERO dataset. To verify the annotation quality, this paper investigates that the existing dialogue model (T5-base) uniformly degrades with the decrease in the amount of available information. Furthermore, the authors propose to apply contrastive learning and this approach helps to fill the information gap.

**Questions For The Authors:**

- What is the motivation behind applying a contrastive learning approach to mitigate the issue of the information gap?
- Do current LLMs truly exhibit a deficiency in their abilities for inductive reasoning?

**Reasons To Accept:**

- This paper provides task difficulty annotation based on the information gap. The annotations will facilitate the research of understanding how the model can perform dialogues with inductive reasoning.
- This paper investigates that existing models struggle with accomplishing inductive reasoning, requiring the ability to understand the information gap in the context of dialogue inference processes.
- Proposed contrastive learning significantly improves the plausibility of the generated inferences.
- Well-written and easy to follow.

**Reasons To Reject:**

- The motivation behind applying a contrastive learning approach to mitigate the issue of the information gap is unclear. The proposed selection of negative samples might not be a straightforward approach to fill the information gap. I think the Non-Optional Generation does not always produce a wrong (negative) output, causing the model to struggle with distinguishing between positive/negative samples.
- While the authors claim the “Current LLMs, such as ChatGPT, lack the called “inductive reasoning” ability” in line 39,  this paper evaluated only LLaMA model as the current LLM in Table 4. Therefore, I didn’t understand how current LLMs can perform inductive reasoning and suggest that other LLMs, such as ChatGPT, might be compared with proposed methods.
- As discussed in Sec. 6.5, I understand evaluating inductive reasoning is challenging. But, I failed to understand that overlap-based metrics (e.g., BLEU) do not reflect its abilities. CICERO Dataset contains many samples requiring inductive reasoning abilities, and therefore I think there is some extent to which overlap-based metrics can be employed to evaluate inference ability.

**Reproducibility:**

5: Could easily reproduce the results.

**Reviewer Confidence:**

4: Quite sure. I tried to check the important points carefully. It's unlikely, though conceivable, that I missed something that should affect my ratings.

**Typos Grammar Style And Presentation Improvements:**

- LLAMA -> LLaMA in Table4 and Table B.1
- 27.34 ROUGE-L score had better the normal font than the bold one in T5-small of Table4

---

> ### Author Rebuttal · Authors · 2023-08-29
>
> Thank you for your comments.
>
> > Weakness 1-1: The motivation behind applying a contrastive learning approach to mitigate the issue of the information gap is unclear. The proposed selection of negative samples might not be a straightforward approach to fill the information gap.
>
> > Question 1: What is the motivation behind applying a contrastive learning approach to mitigate the issue of the information gap?
>
> **Response**: As described in paragraph lines 66-83, contrastive learning is intended to expose negative samples during training to give more guidance. In deductive reasoning, all the information required to generate an output is provided in the input, and there is no information gap. Thus, we may not have to feed negative samples to teach models to reason. However, inductive reasoning requires including something that may not be explicitly stated in the input and that is not simply learnable by only exposing gold samples. Thus, we need to teach the model with more guidance on the reasoning path. In our preliminary experiment using the same dataset and a multi-choice framework, we see a significant improvement by feeding negative samples together with the other candidates, which indicates that feeding negative samples will help the model learn how to fill the information gap.
>
> >Weakness 1-2: I think the Non-Optional Generation does not always produce a wrong (negative) output, causing the model to struggle with distinguishing between positive/negative samples.
>
> **Response**: We use the Non-Optional Generation as one of the models for the ablation study to prove that it is not a proper source for producing negative samples. It is quite noisy and not always wrong/negative, as you have mentioned.
>
> >Weakness 2: While the authors claim the “Current LLMs, such as ChatGPT, lack the called “inductive reasoning” ability” in line 39, this paper evaluated only the LLaMA model as the current LLM in Table 4. Therefore, I didn’t understand how current LLMs can perform inductive reasoning and suggest that other LLMs, such as ChatGPT, might be compared with proposed methods.
>
> >Question 2: Do current LLMs truly exhibit a deficiency in their abilities for inductive reasoning?
>
> **Response**: Inductive reasoning is one of the major weaknesses of LLMs reported and less discussed compared to deductive reasoning ability [Huang and Chang, 2023]. For example, [Laskar et al., 2023] report that even ChatGPT performs much worse in inductive reasoning tasks compared to deductive reasoning tasks.
>
> [Huang and Chang., 2023] Towards Reasoning in Large Language Models: A Survey. ACL2023-Findings.
>
> [Laskar et al., 2023] A Systematic Study and Comprehensive Evaluation of ChatGPT on Benchmark Datasets. ACL 2023.
>
> >Weakness 3: As discussed in Sec. 6.5, I understand evaluating inductive reasoning is challenging. But, I failed to understand that overlap-based metrics (e.g., BLEU) do not reflect its abilities. CICERO Dataset contains many samples requiring inductive reasoning abilities, and therefore I think there is some extent to which overlap-based metrics can be employed to evaluate inference ability.
>
> We agree that automatic metrics can capture a certain degree of quality of generated answers as they align with the human evaluation results (see Table 5 and Section 6.4).
> However, as you can see from the human evaluation results of Ours vs. Gold on the Conceivable level, Ours wins against Gold.
> It is because the conceivable level gold answers tend to include something that is not stated/verifiable in the dialogue contexts provided, while Ours tends to be more supported by the dialogue context (see the examples in Table 1 for a comparison of ours vs. gold and the Conceivable example between line 294-295). In such cases, we believe that solely relying on overlap-based metrics is insufficient.

---

### Official Review · Reviewer_FcSR · 2023-08-10

**Soundness:** 3

**Excitement:**

3: Ambivalent: It has merits (e.g., it reports state-of-the-art results, the idea is nice), but there are key weaknesses (e.g., it describes incremental work), and it can significantly benefit from another round of revision. However, I won't object to accepting it if my co-reviewers champion it.

**Paper Topic And Main Contributions:**

This paper analyzes the models' performance on dialogue understanding with the question-answering task from the aspect of the semantic information gap. They hypothesize that the difficulty of the task is proportional to the amount of this gap, and classify 450 test samples from the CICERO dataset into three levels with human annotators: sufficient, likely, and conceivable, where level 3 "conceivable" should require a higher level of the inductive reasoning ability of models to generate the answer to the corresponding question. The results of a preliminary study with the fine-tuned T5-base models verify the authors' hypothesis.
Then, a contrastive learning approach with different ways for constructing negative samples is proposed to fine-tune the smaller models, such as T5 and GPT2, and achieves consistent improvements with different n-gram-based automatic evaluation metrics and human evaluation on the newly-labeled CICERO test set.

**Questions For The Authors:**

Question A: Is the drop in model performances in Table 2 due to the increases of the information gap or the inaccurate evaluation metrics? It seems that the answer can be more diversified for Conceivable than Sufficient. So, the unrelated but appropriate generated answers compared to the references may show poor performances with those evaluation metrics.

Question B: In Table 3, the subset SE_Clipped is supposed to be more difficult with more Conceivable samples, while its performance is significantly better than SE on CIDEr and Prerequisite on all of the metrics.

Question C: In Table 5, it seems that Ours is more competitive on the Conceivable samples when compared with Gold than the Sufficient samples. Does it contradict the hypothesis? How about the human evaluation result of T5-base vs Gold?

**Reasons To Accept:**

1. The topic of this paper is well-motivated and should be of great interest to the dialogue understanding community.

2. Comprehensive experiments including human evaluation, case study, ablation study, improvements on different question types and task difficulty, etc., are carried out with detailed discussions.

3. The approach is simple and easy to implement.

**Reasons To Reject:**

1. The baselines are too weak. The backbone language models (T5 and GPT2) are also a bit out-of-date. More recent models are recommended, such as T5-Flan, GPT-J, etc. More effective few-shot in-context learning strategies are also expected for the LLAMA baseline, such as selecting the 3 most similar samples instead of random sampling.

2. Some conclusions are inconsistent and confusing. More in Questions.

**Reproducibility:**

4: Could mostly reproduce the results, but there may be some variation because of sample variance or minor variations in their interpretation of the protocol or method.

**Reviewer Confidence:**

4: Quite sure. I tried to check the important points carefully. It's unlikely, though conceivable, that I missed something that should affect my ratings.

---

> ### Author Rebuttal · Authors · 2023-08-29
>
> Thank you for your comments.
>
> > Weakness 1: The baselines are too weak. The backbone language models (T5 and GPT2) are also a bit out-of-date. More recent models are recommended, such as T5-Flan, GPT-J, etc. More effective few-shot in-context learning strategies are also expected for the LLAMA baseline, such as selecting the 3 most similar samples instead of random sampling.
>
> **Response**: First, we have to clarify that we have put LLAMA, the most recent and SOTA LM, as one of our baselines.  In our preliminary experiments with GPT-J, we found its performance is even lower than LLAMA. We also have tried a more strategic prompt using tf-idf to retrieve 3-most similar in-context examples, but it seems that does not help. Note 3-shot are average of three trials with different random seeds. The results are as follows:
>
> | Model |                | BLEU-1 | BLEU-2 | BLEU-3 | BLEU-4 | METEOR | ROUGE_L | CIDEr |
> |-------|----------------|--------|--------|--------|--------|--------|---------|-------|
> | GPT-J | 3-shot         | 24.70  | 13.04  | 6.11   | 3.04   | 13.74  | 25.03   | 19.87 |
> | GPT-J | 3-shot +tf-idf | 22.83  | 11.64  | 5.52   | 2.83   | 12.19  | 22.45   | 17.99 |
> | LLaMA | 3-shot         | 28.43  | 15.18  | 7.25   | 3.73   | 15.26  | 27.43   | 26.47 |
> | LLaMA | 3-shot +tf-idf | 25.36  | 13.33  | 6.56   | 3.49   | 13.72  | 24.91   | 24.06 |
>
> We suspect that even though lexically similar, these examples may mislead the model to make wrong predictions. The reported LLAMA results in this paper are an average of three trials with our well-selected representative examples.
>
> > Weakness 2
>
> > Question A: Is the drop in model performances in Table 2 due to the increases of the information gap or the inaccurate evaluation metrics? It seems that the answer can be more diversified for Conceivable than Sufficient. So, the unrelated but appropriate generated answers compared to the references may show poor performances with those evaluation metrics.
>
> **Response**: Thank you for your question. We believe this diversity in Sufficient samples explains that Ours wins Gold on human evaluation in Table 5 (see the answer to Question C as well). Although automatic metrics are not sufficient, they capture a certain degree of quality of generated answers as they align with the human evaluation results (see Table 5 and Section 6.4).
>
> > Question B:  In Table 3, the subset SE_Clipped is supposed to be more difficult with more Conceivable samples, while its performance is significantly better than SE on CIDEr and Prerequisite on all of the metrics.
>
> **Response**: As described in line 319-329, the proportion of likely and conceivable questions can explain the difference in T5-base performance to a certain extent, it does not have a simple correlation. It may be due to the difference in which kind of information is required to bridge the gap between the dialogue and the answer.  The prerequisite is the most difficult task because the difficulty proportion is more distributed and the model cannot learn the reasoning skill and tendency whether to rely on the dialogue history or extend out of the dialogue history.
>
> > Question C: In Table 5, it seems that Ours is more competitive on the Conceivable samples when compared with Gold than the Sufficient samples. Does it contradict the hypothesis? How about the human evaluation result of T5-base vs Gold?
>
> **Response**: Conceivable level gold answers tend to include something that is not stated/verifiable in the dialogue contexts provided, while Ours tends to be more supported by the dialogue context (see the examples in Table 1 for a comparison of ours vs. gold). We believe this resulted in Ours being more favored by annotators. The sufficient level golds are clearly aligned with dialogue contexts, thus it is more challenging to outperform gold for Ours. We have not conducted the human evaluation of T5-base vs Gold, but will include in the final version of the paper.

---

### Official Review · Reviewer_Qjsz · 2023-08-11

**Soundness:** 3

**Excitement:**

3: Ambivalent: It has merits (e.g., it reports state-of-the-art results, the idea is nice), but there are key weaknesses (e.g., it describes incremental work), and it can significantly benefit from another round of revision. However, I won't object to accepting it if my co-reviewers champion it.

**Paper Topic And Main Contributions:**

This paper studies how language models behave on dialogue inference tasks with different difficulties in terms of information gap, and then uses contrastive learning to improve LMs' performance on the inference task.

The author takes  450 samples from CICERO, a dialogue dataset for commonsense inference, and recruits annotators to classify each sample into three levels of difficulty. The difficulty is defined based on the information gap between the context and target inference, e.g. "sufficient" means all the information for making the inference is presented in the context already. Based on the labels of difficulty, the author shows that LM performs worse for samples with a larger information gap.

The author then proposes to use contrastive learning to improve models' performance on the task. For the CL loss, the objective uses regular in-batch negatives, with the addition of hard negatives from the wrong answers in CICERO datasets, as it is a multiple QA dataset. The author then proposes two other approaches to get hard negatives based on some heuristics, namely non-optional generation based on ideas similar to bootstrapping, and token replacement those tokens affected the most by context are replaced.

The author measured LM's performance with commonly used generation metrics, and added UNLI, a model-based metric for NLI. A human evaluation is carried out and proves the CL method improves more favorable answers over the baseline.


Main Contribution:

1. This work provides a human-annotated dialogue dataset for commonsense inference with task difficulty annotated.

2. The author frames the task difficulty as the information gap and shows that model performance are correlate with an information gap
The author tests contrastive learning on the task with a few ways to get hard negatives and shows that CL increases LMs' performance.

**Questions For The Authors:**

A: Line 442, how is the coefficient chosen? Do you have results from a preliminary experiment?

B: Is table 8 on T5-base, why are BLUE-2 and ROUGE-L for all difficulties higher than the aggregate result?

C: Have you checked for information leakage from the training set to the test set? I'm a bit surprised that T5-small is almost good as T5-large. And what is the prompt you used for LLAMA? you may want to pay more attention to the prompt, as LLAMA is not instruction fine-tuned.


**Reasons To Accept:**

1. The framing of the question is very interesting. Based on the observation that LMs do worse on inductive reasoning, the author attempts to connect that to differences in task difficulty based on information gap. This framing can provide some insights into issues faced by LLMs, for example, if we find the response from LLM has a large information gap, we may want to be more cautious and introduce extra guardrails, etc.  So overall I would like to see more analysis from the community on this.

2. A dataset is presented with annotations of task difficulty for commonsense inference. It has 450 samples which is a decent number, and can be useful.

3. The author tested CL on the task, and show that it improves LMs' performance both in automatic metrics and human evaluation.

**Reasons To Reject:**

1. My main complaint is that the two topics of this work, information gap and contrastive learning, do not glue together very well. Although results for different difficulties were shown in Table 5 and Table 8, the CL method itself has nothing to do with the discussion of inductive vs. deductive, and is not using the information gap labels. That is to say, the CL part is using the standard CL method, plus two heuristics of getting negative samples, in a task-agnostic way on this specific dataset, which offers limited new insight to the community.
I would encourage the author to look a bit more into how will the CL algorithm be different if we know what is the information gap for each sample. For example, maybe samples with a larger information gap are more difficult and need more help from the negative samples, while the "sufficient" ones are easy enough so some computation can be pruned during the training.

2. There are several claims made without sufficient analysis. For line 322 and the examples followed, if the emotion is easily guessed by the sentiment, then that should reflect in the annotations as well. The argument in the paper explains the distribution of difficulty, NOT how it relates to the automatic metrics.

Line 522, for "non-optimal" and "Replace ZS", I don't think they follows the claim of "have a positive impact", and for the "Replace MCQ", we would expect a statistic significance before making the claim

Line 562, "promote the model to conduct inference either deductively or inductively", there is no proof and no example for this claim, and feels like it comes out from nowhere.

Line 459, "the UNLI scores are hardly interpretable given SNLI". NO, the number listed suggests it works just as intended on SNLI. Consider replacing the claim with something from section 6.5 instead, which is a section that is worth more emphasis.

3. Some results seem to be an error either from the write-up or experiment. Compare the BLEU-2 and ROUGE-L columns in  Table 8 with Table 4, It is not possible that each difficulty has higher metrics than the aggregate results through average.

4. A few mistakes in the formulation in section 4, line 333, all the j should be replaced with "i". Line 337, according to the description, the negative set is different for each utterance, either remove the SUM on "n" and state it is the loss for an utterance, or add superscript "n" to A as done with A_tilt.

**Reproducibility:**

4: Could mostly reproduce the results, but there may be some variation because of sample variance or minor variations in their interpretation of the protocol or method.

**Reviewer Confidence:**

5: Positive that my evaluation is correct. I read the paper very carefully and I am very familiar with related work.

**Typos Grammar Style And Presentation Improvements:**

line 26, "go beyond" should be "goes beyond".

Given sec6.5, maybe consider removing UNLI if you think it doesn't really make sense.

For table3, maybe look a bit more into the performance of samples with the same difficult level but with different relationships, before making the claim in line 322

---

> ### Author Rebuttal · Authors · 2023-08-29
>
> Thank you for your comments.
>
> > Weakness 1-1: My main complaint is that the two topics of this work, information gap and contrastive learning, do not glue together very well. Although results for different difficulties were shown in Table 5 and Table 8, the CL method itself has nothing to do with the discussion of inductive vs. deductive, and is not using the information gap labels. That is to say, the CL part is using the standard CL method, plus two heuristics of getting negative samples, in a task-agnostic way on this specific dataset, which offers limited new insight to the community.
>
> **Response**: As described in paragraph lines 66-83, contrastive learning is intended to expose negative samples during training to give more guidance. In deductive reasoning, all the information required to generate an output is provided in the input, and there is no information gap. However, inductive reasoning requires including something that may not be explicitly stated in the input, and that is not simply learnable by only exposing gold samples. Thus, we need to teach the model with more guidance on the reasoning path. In our preliminary experiment using the same dataset and a multi-choice framework, we see a significant improvement by feeding negative samples together with the other candidates, which indicates that feeding negative samples will help the model learn how to fill the information gap.
>
> > Weakness 1-2: I would encourage the author to look a bit more into how will the CL algorithm be different if we know what is the information gap for each sample. For example, maybe samples with a larger information gap are more difficult and need more help from the negative samples, while the "sufficient" ones are easy enough so some computation can be pruned during the training.
>
> **Response**: This is a great point and a really interesting idea, however, this requires an expensive annotation over all the samples in the train set. As discussed in the paper, there is no automatic approximation method to estimate information gaps, and thus applying such techniques is not realistic. In the final version of the paper, we will add an analysis of the impact of the number of negative samples during training, observing the performances on the different levels.
>
> > Weakness 2-1: There are several claims made without sufficient analysis. For line 322 and the examples followed, if the emotion is easily guessed by the sentiment, then that should reflect in the annotations as well. The argument in the paper explains the distribution of difficulty, NOT how it relates to the automatic metrics.
>
> **Response**: In Table 2 and Table 3, we reported the breakdown of the automatic evaluation metrics based on the levels so that we support our claim that “information gap can explain the difficulty in inference in dialogue”. For line 322 and the examples followed, we wanted to point out that the poorly correlated pair between difficulty levels and automatic metrics is caused by another aspect, which is the type of information gap.
>
> > Weakness 2-2: Line 522, for "non-optimal" and "Replace ZS", I don't think they follows the claim of "have a positive impact", and for the "Replace MCQ", we would expect a statistic significance before making the claim
>
> **Response**: Thank you for your clarification. What we meant here is that negative samples in relatively good quality (Replace_MCQ and contradiction) have a positive impact on the performance, but more noisy negative samples are less helpful.
>
> > Weakness 2-3: Line 562, "promote the model to conduct inference either deductively or inductively", there is no proof and no example for this claim, and feels like it comes out from nowhere.
>
> **Response**: “Deductively” and “inductively” correspond to “sufficient” level and “conceivable” level, respectively.  We made this claim based on the human judgment of plausibility presented in Table 5. Please refer to Figure A.1 for annotator instructions to see how they correlate to these levels. For examples to support this claim, please refer to Table 1, B.2, and B.3, where we present dialogues, questions, and gold/T5-base/Ours for comparisons on inference ability. Given your comments, we would like to add more explanations when we introduce the difficulty levels.
>
> > Weakness 2-4: Line 459, "the UNLI scores are hardly interpretable given SNLI". NO, the number listed suggests it works just as intended on SNLI. Consider replacing the claim with something from section 6.5 instead, which is a section that is worth more emphasis.
>
> **Response**: Thank you for your suggestion, we will replace the claim and reorganize the corresponding part.
>
> > Weakness 3: Some results seem to be an error either from the write-up or experiment. Compare the BLEU-2 and ROUGE-L columns in Table 8 with Table 4, It is not possible that each difficulty has higher metrics than the aggregate results through average.
>
> > Question B: Is table 8 on T5-base, why are BLUE-2 and ROUGE-L for all difficulties higher than the aggregate result?
>
> **Response**: This is because Table 8 reports T5-base with contrastive learning scores on the subset of the test set that has task difficulty annotation, resulting in different numbers reported in Table 4. The scores of T5-base without contrastive learning in the same subset are reported in Table 2, and the comparisons are made against Table 2.
>
> > Weakness 4: A few mistakes in the formulation in section 4, line 333, all the j should be replaced with "i". Line 337, according to the description, the negative set is different for each utterance, either remove the SUM on "n" and state it is the loss for an utterance, or add superscript "n" to A as done with A_tilt.
>
> **Response**: Thank you so much for the correction! We will update correspondingly in the final version.
>
> > Question A: Line 442, how is the coefficient chosen? Do you have results from a preliminary experiment?
>
> **Response**: We test with several coefficients and report the one with the best results in the paper. The results from the preliminary experiment with T5-base are shown below:
>
> | lambda_b | lambda_s | BLEU-1 | BLEU-2 | BLEU-3 | BLEU-4 | METEOR | ROUGE_L | CIDEr |
> |----------|----------|--------|--------|--------|--------|--------|---------|-------|
> | 0.5      | 0.5      | 30.67  | 17.09  | 9.45   | 5.65   | 16.62  | 28.50   | 40.53 |
> | 1.0      | 1.0      | 30.29  | 16.69  | 9.09   | 5.42   | 16.55  | 28.31   | 38.77 |
> | 2.0      | 2.0      | 29.81  | 16.37  | 8.88   | 5.28   | 16.32  | 28.05   | 38.18 |
>
> > Question C-1: Have you checked for information leakage from the training set to the test set? I'm a bit surprised that T5-small is almost good as T5-large. And what is the prompt you used for LLAMA? you may want to pay more attention to the prompt, as LLAMA is not instruction fine-tuned.
>
> **Response**: Regarding the information leakage from the training set to the test set following [Elangovan et al., 2021]. For each sample in the test set, the similarity between all the samples is evaluated and the highest similarity sample is chosen. Then, we take the average of the similarity over all the test instances as an indicator to measure the extent of train/test overlap. We compute their overlap by representing instances with a count vector of uni/bi/trigrams and computing the cosine similarity. We compare the similarity between dialogues and answers in the train set and test set. The results are as follows:
>
> |          | unigram (%) | bigram (%) | trigram (%) |
> |----------|-------------|------------|-------------|
> | dialogue | 68.54       | 20.15      | 9.67        |
> | answer   | 59.91       | 35.62      | 13.31       |
>
> Considering that 60% is the threshold of being “highly overlapped” [Elangovan et al., 2021], the dialogue and answers in the CICERO train and test set have some level of overlap between them in unigrams. However, the overlap does not necessarily mean leakage. For example, a generic answer "the speaker is satisfied with the listener's proposal" can appear in both train/test sets.
>
> [Elangovan et al., 2021] Memorization vs. Generalization: Quantifying Data Leakage in NLP Performance Evaluation. EACL 2021.
>
> >Question C-2:  And what is the prompt you used for LLAMA? you may want to pay more attention to the prompt, as LLAMA is not instruction fine-tuned.
>
> **Response**: You can find prompts used in the supplemental material. In our preliminary experiments, we tried different prompts and retrieved 3-most similar samples to input, but it did not help. In the paper, we put the best results of LLAMA. Please refer to the answer to Reviewer FcSR Weakness 1.
>
> > Suggestions: For table3, maybe look a bit more into the performance of samples with the same difficult level but with different relationships, before making the claim in line 322
>
> **Response**: Thank you for your suggestion. Due to the page limit, we have put the breakdown scores for each relation in Table B.1 in the appendix. We will clearly point it out in the camera-ready.

---

### Official Review · Reviewer_X1eN · 2023-08-11

**Soundness:** 3

**Excitement:**

3: Ambivalent: It has merits (e.g., it reports state-of-the-art results, the idea is nice), but there are key weaknesses (e.g., it describes incremental work), and it can significantly benefit from another round of revision. However, I won't object to accepting it if my co-reviewers champion it.

**Paper Topic And Main Contributions:**

The paper proposes a contrastive learning approach to improve LLM's capability of inductive reasoning.

The authors first demonstrate that a finetuned baseline T5 model performs worse when the information gap for the inference task is bigger. Then they propose a contrastive learning approach with different variants in negative samples selection. The authors evaluate the method in dataset CICERO using automatic metrics and human evaluations.

**Questions For The Authors:**

(A) Based on human evaluation, the proposed method's performance is better than gold. Do you have more thorough analysis on it to learn where the gain is from?

**Reasons To Accept:**

- The idea behind the proposed method is intuitive and simple (which is good) and the method is model architecture-agnostic.

- The problem the paper tries to address, Improving LLM's capability of inductive reasoning, is an important research topic for LLM, especially now LLMs, like ChatGPT, are becoming increasingly popular in the public eye.

- The paper clearly states the proposed method with sufficient details and has done thorough evaluation and ablation study to help readers to gain a thorough understanding.

**Reasons To Reject:**

- The results of different metrics are not telling a consistent story, as the authors claimed in the paper.
  - (Table 4) The proposed method (+CL) shows improvements in n-gram overlap-based evaluation metrics across all test models, but NLI based metrics produce mixed results: it seems there's only improvements in GPT2-base model, while no improvement (some even with performance hurt) in T5 models.
  - Based on overlap-based evaluation metrics (Table 8), the authors reach to the conclusion that the method "promote the model to conduct inference either fully deductively or inductively", while based on human evaluation (Table A.2), the improvement is only significant in "Sufficient" level.

- As the authors also mentioned, the paper only has experiment results on one dataset. It'd be helpful to test the method on at least another one.



**Reproducibility:**

4: Could mostly reproduce the results, but there may be some variation because of sample variance or minor variations in their interpretation of the protocol or method.

**Reviewer Confidence:**

3: Pretty sure, but there's a chance I missed something. Although I have a good feel for this area in general, I did not carefully check the paper's details, e.g., the math, experimental design, or novelty.

---

> ### Author Rebuttal · Authors · 2023-08-29
>
> Thank you for your comments.
>
> > Weakness 1-1: (Table 4) The proposed method (+CL) shows improvements in n-gram overlap-based evaluation metrics across all test models, but NLI based metrics produce mixed results: it seems there's only improvements in GPT2-base model, while no improvement (some even with performance hurt) in T5 models.
>
> **Response**: We have already included a discussion about NLI metrics in Section 6.5, and our results are consistent. The reason why we put the results on the UNLI metric is not to evaluate the performance of our proposed models, but to analyze whether it is a feasible metric for commonsense reasoning or not. We conclude that NLI metrics cannot evaluate the reasoning ability of the generated texts, since most of the time gold answers are labeled as “neutral” and undeterminable, and it is extremely difficult to associate the numbers with the quality of generated inferences. To mitigate the potential confusion, we put this information separately in the final version of the paper.
>
> > Weakness 1-2: Based on overlap-based evaluation metrics (Table 8), the authors reach to the conclusion that the method "promote the model to conduct inference either fully deductively or inductively", while based on human evaluation (Table A.2), the improvement is only significant in "Sufficient" level.
>
> **Response**: In Table A.2, ours is significantly better than t5-base at the sufficient level, and than gold at the conceivable level.
> Table 5 clearly shows that even though not significant at the Conceivable level, a majority of annotations assess our model's wins over the T5-base in both Sufficient and Conceivable levels, which aligns with the overlap-based results. In Table 8, we show that our method improves the performance more at sufficient and conceivable levels.
> Thus, we concluded that our method promotes the model to conduct inference either fully deductively or inductively.
>
> > Question-A: Based on human evaluation, the proposed method's performance is better than gold. Do you have more thorough analysis on it to learn where the gain is from?
>
> **Response**: Based on the insights gained from the human evaluation (see Table 5), our proposed method is NOT better than gold, but it is comparable to gold, except at the Conceivable level. Conceivable level gold answers tend to include something that is not stated/verifiable in the dialogue contexts provided, while Ours tends to be more supported by the dialogue context (see the examples in Table 1 for a comparison of ours vs. gold). We believe this resulted in Ours being more favored by annotators.

---

### Official Review · Reviewer_eCnU · 2023-08-14

**Soundness:** 3

**Excitement:**

3: Ambivalent: It has merits (e.g., it reports state-of-the-art results, the idea is nice), but there are key weaknesses (e.g., it describes incremental work), and it can significantly benefit from another round of revision. However, I won't object to accepting it if my co-reviewers champion it.

**Paper Topic And Main Contributions:**

The paper tries to analyze the problems in the inductive reasoning ability of the language models as a function of semantic information gap present in the language task model is trying to solve. They also present a way to improve the model performance and empirically show that their method is effective in many cases.

**Questions For The Authors:**

One comment I would like to make is regarding the choice of automated evaluation metrics, themselves. The authors mainly show improvements using text similarity metrics which seem like a poor choice to judge inductive reasoning capabilities of a model.

**Reasons To Accept:**

1- The authors present a way to improve the inductive reasoning capability language models through contrastive learning.
2- Authors formalize the notion of semantic information gap and empirically show that their approach improves the model's performance in both human and automatic evaluations.

**Reasons To Reject:**

1- The average performance improvements on the CICERO test-set are minor. Similarly, an analysis of why NLI metrics don't improve would have been welcome.

**Reproducibility:**

3: Could reproduce the results with some difficulty. The settings of parameters are underspecified or subjectively determined; the training/evaluation data are not widely available.

**Reviewer Confidence:**

4: Quite sure. I tried to check the important points carefully. It's unlikely, though conceivable, that I missed something that should affect my ratings.

**Typos Grammar Style And Presentation Improvements:**

small spelling error:
line 193: lake -> lack

---

> ### Author Rebuttal · Authors · 2023-08-29
>
> Thank you for your comments.
>
> >Weakness 1-1: The average performance improvements on the CICERO test set are minor.
>
>
> **Response**: As shown in Table 5, although the improvements on the CICERO test set (Table 4) are minor on automatic metrics, our method is significantly better based on human evaluation.
>
>
> >Weakness 1-2: Similarly, an analysis of why NLI metrics don't improve would have been welcome.
>
> **Response**: We have included a discussion about NLI metrics in Section 6.5. The reason why we put the results on the UNLI metric is not to evaluate the performance of our proposed models, but to analyze whether it is a feasible metric for commonsense reasoning or not. To mitigate the potential confusion, we will put this information separately in the final version of the paper in section 6.5.
>
>
> >Question 1: One comment I would like to make is regarding the choice of automated evaluation metrics, themselves. The authors mainly show improvements using text similarity metrics which seem like a poor choice to judge inductive reasoning capabilities of a model.
>
> **Response**: We agree that showing improvements using only text similarity metrics is a poor choice to judge the inductive reasoning capabilities of a model.
> To further evaluate the performance, we provided a comprehensive human evaluation in Section 5.2 and our method achieved significant improvement over the baseline.
>
> >Typos etc.
>
> **Response**: Thank you for your correction. We will modify them accordingly.

---

### Meta-Review · Area_Chair_sweW · 2023-09-19

**Recommendation:** 3

**Metareview:**

The paper examines the inductive reasoning ability of language models in dialogue-based tasks by analyzing the semantic information gap between the context and the inference required. The authors propose a contrastive learning approach to improve model performance. They conduct thorough experiments and evaluations, including human evaluation, to support their claims.

The reviewers generally find the topic interesting and well-motivated. They appreciate the clarity and thoroughness of the experiments and analysis. The proposed method is considered simple and easy to implement.

However, there are some concerns raised by the reviewers. One reviewer points out that the average performance improvements on the test set are minor and suggests analyzing why NLI metrics don't improve. Another reviewer questions the consistency of the results across different metrics and asks for further analysis. There are also remarks about inconsistent conclusions, weak baselines, and the need for more recent models and few-shot learning strategies.

In terms of the contributions, the human-annotated dialogue dataset for commonsense inference task difficulty classification is considered valuable. The framing of the research question is also seen as interesting. The contrastive learning approach shows consistent improvements.

Overall, while there are some weaknesses and areas for improvement, the paper provides sufficient support for its major claims. The reviewers have mixed levels of excitement, with some suggesting further revisions. But none of them explicitly object to accepting the paper if co-reviewers support it.

In summary, the paper addresses an important research topic, presents a valuable dataset, and proposes a promising method to improve language model performance in dialogue-based tasks. Some revisions and clarifications are recommended to address the concerns raised by the reviewers.

---

### Decision · Program_Chairs · 2023-10-07

**Decision:**

Accept-Main

**Comment:**

The paper examines the inductive reasoning ability of language models in dialogue-based tasks by analyzing the semantic information gap between the context and the inference required. The authors propose a contrastive learning approach to improve model performance. They conduct thorough experiments and evaluations, including human evaluation, to support their claims.

The reviewers generally find the topic interesting and well-motivated. They appreciate the clarity and thoroughness of the experiments and analysis. The proposed method is considered simple and easy to implement.

However, there are some concerns raised by the reviewers. One reviewer points out that the average performance improvements on the test set are minor and suggests analyzing why NLI metrics don't improve. Another reviewer questions the consistency of the results across different metrics and asks for further analysis. There are also remarks about inconsistent conclusions, weak baselines, and the need for more recent models and few-shot learning strategies.

In terms of the contributions, the human-annotated dialogue dataset for commonsense inference task difficulty classification is considered valuable. The framing of the research question is also seen as interesting. The contrastive learning approach shows consistent improvements.

Overall, while there are some weaknesses and areas for improvement, the paper provides sufficient support for its major claims. The reviewers have mixed levels of excitement, with some suggesting further revisions. But none of them explicitly object to accepting the paper if co-reviewers support it.

In summary, the paper addresses an important research topic, presents a valuable dataset, and proposes a promising method to improve language model performance in dialogue-based tasks. Some revisions and clarifications are recommended to address the concerns raised by the reviewers.